# The Role of Chest CT Radiomics in Diagnosis of Lung Cancer or Tuberculosis: A Pilot Study

**DOI:** 10.3390/diagnostics12030739

**Published:** 2022-03-18

**Authors:** Lekshmi Thattaamuriyil Padmakumari, Gisella Guido, Damiano Caruso, Ilaria Nacci, Antonella Del Gaudio, Marta Zerunian, Michela Polici, Renuka Gopalakrishnan, Aziz Kallikunnel Sayed Mohamed, Domenico De Santis, Andrea Laghi, Dania Cioni, Emanuele Neri

**Affiliations:** 1Department of Radiology, Apollo Adlux Hospital, Kochi 683576, Kerala, India; drlekshmitp@gmail.com; 2Department of Medical Surgical Sciences and Translational Medicine, Sapienza University of Rome, Radiology Unit—Sant’Andrea University Hospital, Via di Grottarossa, 1035, 00189 Rome, Italy; gisella.guido@uniroma1.it (G.G.); ilaria.nacci@uniroma1.it (I.N.); antonella.delgaudio@uniroma1.it (A.D.G.); marta.zerunian@uniroma1.it (M.Z.); michela.polici@uniroma1.it (M.P.); domenico.desantis@hotmail.it (D.D.S.); andrea.laghi@uniroma1.it (A.L.); 3Department of Radiodiagnosis, Regional Cancer Centre, Thiruvananthapuram 695011, Kerala, India; drrenukag@gmail.com; 4Department of Pulmonology, Apollo Adlux Hospital, Kochi 683576, Kerala, India; drazizksm@gmail.com; 5Italian Society of Medical and Interventional Radiology, SIRM Foundation, Via della Signora 2, 20122 Milano, Italy; dania.cioni@unipi.it (D.C.); emanuele.neri@unipi.it (E.N.); 6Academic Radiology, Department of Translational Research, University of Pisa, Via Roma, 67, 56126 Pisa, Italy

**Keywords:** chest CT, radiomics, lung cancer, tuberculosis, multidetector computed tomography, texture analysis, oncology, precision medicine, lung imaging

## Abstract

In many low-income countries, the poor availability of lung biopsy leads to delayed diagnosis of lung cancer (LC), which can appear radiologically similar to tuberculosis (TB). To assess the ability of CT Radiomics in differentiating between TB and LC, and to evaluate the potential predictive role of clinical parameters, from March 2020 to September 2021, patients with histological diagnosis of TB or LC underwent chest CT evaluation and were retrospectively enrolled. Exclusion criteria were: availability of only enhanced CT scans, previous lung surgery and significant CT motion artefacts. After manual 3D segmentation of enhanced CT, two radiologists, in consensus, extracted and compared radiomics features (T-test or Mann–Whitney), and they tested their performance, in differentiating LC from TB, via Receiver operating characteristic (ROC) curves. Forty patients (28 LC and 12 TB) were finally enrolled, and 31 were male, with a mean age of 59 ± 13 years. Significant differences were found in normal WBC count (*p* < 0.019) and age (*p* < 0.001), in favor of the LC group (89% vs. 58%) and with an older population in LC group, respectively. Significant differences were found in 16/107 radiomic features (all *p* < 0.05). LargeDependenceEmphasis and LargeAreaLowGrayLevelEmphasis showed the best performance in discriminating LC from TB, (AUC: 0.92, sensitivity: 85.7%, specificity: 91.7%, *p* < 0.0001; AUC: 0.92, sensitivity: 75%, specificity: 100%, *p* < 0.0001, respectively). Radiomics may be a non-invasive imaging tool in many poor nations, for differentiating LC from TB, with a pivotal role in improving oncological patients’ management; however, future prospective studies will be necessary to validate these initial findings.

## 1. Introduction

Tuberculosis (TB) and lung cancer (LC) are the most common lung diseases contributing to mortality in developing nations, and both have associations with the human development index of the country or region [1]. LC is the second most common subsite, contributing 11.4% of new cases and 18% of new deaths, according to the Global cancer statistics 2020-GLOBOCAN estimates [2].

LC and TB are often known to coexist. In fact, some studies have shown chronic inflammation of TB as carcinogenic [3,4]. In previous studies, the coexistence of TB and LC, in a small percentage of patients, was documented [5,6,7]. Furthermore, lung malignancy and the different drugs used for the treatment are associated with immunosuppression, which often leads to mycobacterial infection [8,9].

Due to the high prevalence of TB in endemic areas, lung consolidation is presumptively treated with antibiotics and antituberculosis drugs with a wait and watch policy [10]. The lack of response to treatment is considered an indication to suspect malignancy, and thus, the treatment may be delayed [11]. This often results in LC presenting at an advanced stage, with a dramatic impact on the survival [12]. The limited access to pulmonary biopsy usually results in the upstaging of the malignancy or, in uncertain diagnoses or a watch and wait strategy, results in increased mortality and morbidity [13].

Imaging examinations, such as computed tomography (CT), demonstrated key tools in differential diagnosis between TB and LC [14]. Nevertheless, in clinical practice, due to the radiological similarities between these pathologies, even expert radiologists, relying on CT data, are often subject to misdiagnosis [15]. In particular, the presence of chest CT findings, as consolidation and a nodular pattern, especially those with spiculations and irregular margins in many tuberculous bacteria, can mimic primary LC [16]. As a corollary finding, in a case of known malignancy, a tubercular non-calcified granuloma is sometimes misdiagnosed as a metastatic nodule [17]. Thus, a non-invasive and diagnostic alternative is required to improve the discrimination between TB and LC.

In this contest, Radiomics is an emerging medical imaging tool that turns the qualitative analysis of multimodal medical images into quantitative data [18,19]. After imaging feature extraction with dedicated software, trained algorithms provide diagnostic aid or exact quantitative information by calculating the extracted features [20]. Thus, Radiomics can reflect biological information regarding the analyzed lung lesions, such as cell morphology, internal heterogeneity, and molecular and gene expression, which can provide a more accurate differential diagnosis for confused masses, in a non-invasive way [21,22,23].

In recent literature, only a few studies have investigated the role of quantitative radiomic features to differentiate lung TB from LC [24,25,26].

The aim of the study was to identify and compare the CT radiomic features of both TB and LC, as well as identify the best ones, in order to demonstrate the potential key role of radiomics in differentiating between these two diseases, allowing the accurate assessment and early diagnosis of LC in developing nations.

## 2. Materials and Methods

### 2.1. Patient Population and Study Design

This retrospective observational study was in accordance with the Declaration of Helsinki. All participants provided informed consent, and the approval of the Institutional Review Board was not necessary in this observational non-interventional retrospective study.

Sixty patients admitted at Apollo Adlux Hospital, Kerala, India, from March 2020 to September 2021, were selected according to the following inclusion criteria: patients with (a) histological diagnosis of TB or LC, as well as patients (b) who underwent chest CT during hospitalization.

Exclusion criteria were: (a) negative chest CT for pulmonary consolidation, (b) availability of the only CT scans with contrast media administration, (c) previous surgical pulmonary resection, and (d) significant motion artefacts on chest CT.

Patients’ demographic characteristics, clinical findings, and laboratory results, including sex, age, comorbidities, smoking habits, respiratory symptoms, fever, erythrocyte sedimentation ratio (ESR), white blood cell (WBC) count, hemoglobin (HB), and histopathology results, were also retrieved from the internal hospital records and analyzed.

### 2.2. CT Acquisition Technique

All patients underwent unenhanced Chest CT scans during hospitalization. Chest CT acquisitions were obtained with the patients in supine position during end-inspiration, without contrast medium injection, and with the scans performed in the cranio-caudal direction. CT exams were obtained by using 160-slice CT (CANON Aquilion SP Prime Scanner, Canon Medical Systems Corporation, Otawara, Japan). CT scans were obtained by setting the following technical parameters: tube voltage 120 kV; tube current modulation 300 mAs, spiral pitch factor 0.98; collimation 64 × 0.625 mm; time of rotation 0.5 s. Standard soft tissue reconstruction, by Iterative Reconstruction, was used for all CT images at a slice thickness of 0.5 mm.

### 2.3. CT Scans Evaluation and Segmentation Analysis

Digital Imaging and Communications in Medicine (DICOM) data were transferred into a picture archiving and communication system (PACS) workstation (Centricity Universal Viewer, version 6.0; GE Medical Systems, Boston, Massachussets, United States). Two radiologists, in consensus (G.G. and D.C., with 5 years and 15 years in thoracic imaging experience, respectively), evaluated the CT scans eligible for segmentation analysis; they then performed CT scan segmentation analysis. The volumetric lung segmentation of each CT scan was performed by using open-source 3D Slicer software (version 4.11.20210226, http://www.slicer.org, accessed on 28 February 2021). Slice-by-slice, a volumetric region of interest (VOI) was manually drawn on mediastinal window scans, with the goal of covering total consolidation volume and avoiding the pulmonary vessels, or bronchi, and cavitations.

### 2.4. Radiomic Features Extraction

The 3D Slicer Radiomics extension (pyradiomics library [18]) was used to extract 107 radiomic features from the mediastinal window of unenhanced Chest CT scans, including first and second order features: 19 features first order statistics, 13 features 2D and 3D shapes, 16 features Grey Level Size Zone Matrix (GLSZM), 5 features Neighbouring Gray Tone Difference Matrix (NGTDM), 14 features Gray Level Dependence Matrix (GLDM), 24 features gray-level co-occurrence matrix (GLCM), and 16 features gray level run length matrix (GLRLM).

### 2.5. Statistical Analysis

All data are expressed as mean ± standard deviation (SD). Categorical variables were described as counts and percentages. Gaussian distribution was tested by the Shapiro–Wilk test: continuous parametric variables were compared by using the Student *t*-test, while non-parametric variables were compared with the Mann–Whitney U test. Statistical significance was assessed with *p* < 0.05. For inferential comparisons, correction for multiple testing was done with the Holm–Bonferroni method; that is, the smallest *p* value was compared to 0.05/107 = 0.00047 [27]. Statistical analysis was performed using MedCalc Statistical Software version 20.013 (MedCalc Software bvba, Ostend, Belgium).

Receiver operating characteristic (ROC) curves, and the calculated areas under the curve (AUCs), were calculated to test the significant performance of chest CT radiomic features in differentiating LC from TB; sensitivity and specificity were evaluated too.

## 3. Results

### 3.1. Study Population and Patients Data

From an initial population of 60 patients, ten (16%) were excluded for the absence of TB consolidation at chest CT examination, three (5%) were for previous pulmonary surgery, two (3%) were for the availability of only CT scans with contrast media administration, and five (8%) patients were excluded for the presence of severe motion artifacts on chest CT images.

Thus, the final population comprised forty patients who were finally enrolled, where 31 were male (77%) and 9 were female (23%), with a mean age of 59 ± 13 (SD) years and an age range of 21–82. Among those, 28 patients (70%) were affected by LC, and 12 patients (30%) were affected by TB. The enrollment flowchart of the study is shown in Figure 1.

There were 19 (48%) patients who had a current smoking habit, and 21 (52%) had never smoked. Additionally, 29 (73%) participants had at least one underlying comorbidity, and diabetes mellitus and hypertension were the most common reported. Among the patients, 33 (82%) reported symptoms such as cough (16/40), dyspnea (7/40), fever (11/40), and weight-loss (9/40), while seven patients (18%) were asymptomatic. The correlation of patients’ data and clinical parameters, between the two patient groups, showed that significant differences were found in normal WBC count (*p* < 0.019) and age (*p* < 0.001), in favor of the LC group (89% vs. 58%) and with an older population in LC group, respectively. No other statistically significant differences were found, despite LC patients presenting cough more than the TB group (13 vs. 3).

Full data about patients’ demographics, clinical records, and laboratory findings are reported in Table 1.

### 3.2. 3D Segmentation and Radiomic Features

Figure 2 and Figure 3 show CT scan examples of LC and TB; volumetric segmentations of both LC and TB are shown in Figure 4 and Figure 5.

From the volumetric segmentation of lung parenchyma, the software itself has routinely extracted 107 radiomic features from chest CT scans. In the comparison between LC patients and TB patients, 16 radiomic parameters showed significantly different results after correction for multiple testing (Table 2).

In particular, among Shape features, only Surface Volume Ratio was able to significantly differentiate between two patient groups (*p* = 0.0003) after adjustment for multiple testing, with a good performance (AUC: 0.868, sensitivity: 85.7%, specificity: 83.3%, *p* < 0.0001) in discriminating LC patients.

Among First Order features, two (10Percentile and Mean) significantly differentiated LC and TB patients (*p* = 0.0002 and *p* =0.0003 respectively); 10Percentile showed the best performance (AUC: 0.881, sensitivity: 75%, specificity: 91.7%, *p* <0.0001) for LC patients’ individuation.

Significant differences were found in two Grey Level Co-occurrence Matrix (GLCM) features, (Difference Average and Idm) which demonstrated significant differences between two groups (both *p* = 0.0004) with the best performance of Idm (AUC: 0.857, sensitivity: 92.9%, specificity: 66.7%, *p* < 0.0001) in discriminating LC. 

Among Gray Level Dependence Matrix (GLDM) features, two (Large Dependence Emphasis, and Small Dependence Emphasis,) were able to differentiate, in a significant way, between TB and LC (*p* = 0.00000026–0.000016), with the best performance of Large Dependence Emphasis (AUC: 0.92, sensitivity: 85.7%, specificity: 91.7%, *p* < 0.0001) in discriminating LC.

Significant differences were found in five Grey-Level Run Length Matrix (GLRLM) features, (Long Run Emphasis, Run Length Non Uniformity Normalized, Run Percentage, Run Variance, Short Run Emphasis), (*p* < 0.0000077–0.000028); among them, Run Variance showed the best performance (AUC: 0.938, sensitivity: 85.7%, specificity: 100%, *p* < 0.0001) to distinguish LC from TB.

Among Gray Level Size Zone Matrix (GLSZM) features, four (Large Area Emphasis, Large Area Low Gray Level Emphasis, Zone Percentage, and Zone Variance) showed significant differences between TB and LC (*p* < 0.000017–0.0002), and Large Area Low Gray Level Emphasis demonstrated the best performance in discriminating LC (AUC: 0.92, sensitivity: 75%, specificity: 100%, *p* < 0.0001).

After adjustment for multiple testing, no Neighboring Gray Tone Difference Matrix (NGTDM) features were able to discriminate, in a significant way, between LC and TB group. Figure 6 shows the best AUCs for each category of radiomic features.

## 4. Discussion

This study investigated radiomic features of pulmonary tuberculosis (TB) and lung cancer (LC), to assess the potential role of Radiomic in differentiating between these two diseases, in 40 patients living in developing country. Significant differences were found after adjustment for multiple testing in 16/107 radiomic features extracted: 1 Shape, 2 First Order, 2 GLCM, 2 GLDM, 5 GLRLM, and 4 GLSZM, all with *p* < 0.05. Furthermore, Large Dependence Emphasis (GLDM feature) and Large Area Low Gray Level Emphasis (GLSZM feature) showed the best performance in discriminating LC from TB (AUC: 0.92, sensitivity: 85.7%, specificity: 91.7%, *p* < 0.0001; AUC: 0.92, sensitivity: 75%, specificity: 100%, *p* < 0.0001, respectively). The correlation of patients’ data and clinical parameters showed significant differences in normal WBC count (*p* < 0.019) and age (*p* < 0.001), with an older population in the LC group.

Our results, in line with previous literature studies, reinforced the idea that Radiomics could have a future role in the management of patients affected by pulmonary mass, in particular in low-income nations, to reach the diagnosis earlier and to avoid treatment’s delay.

To date, only a few studies investigated the role of Radiomics in differential diagnosis between LC and TB. In particular, E.N. Cui and colleagues [25] validated a radiomics method for distinguishing pulmonary TB from LC, based on CT images, by analyzing peritumoral regions with good discrimination (AUC: 0.91 for training cohort, and 0.90 for validation cohort).

In a remarkable study conducted by Feng B. et al. [26], in line with our results, 426 patients were enrolled to investigate the radiomics nomogram’s differential diagnostic performance in discriminating between tuberculous granuloma and lung adenocarcinoma, appearing as solitary pulmonary solid nodules. Individualized radiomics nomograms, incorporating the radiomics features and clinical factors, were constructed to validate the diagnostic ability. Radiomics signature, age, and spiculation sign, being independent predictors, were used to build the radiomics nomogram, which showed higher diagnostic accuracy than each single model (AUCs: 0.966, 0.934, and 0.906 for training, internal validation, and external validation cohorts, respectively).

Similarly, some studies have focused on the Radiomics performance to discriminate between LC and atypical granulomas. In detail, Yang X. et al. investigated the ability of quantitative CT radiomics to preoperatively differentiate solitary atypical granulomatous nodules from lung adenocarcinoma, in 302 patients, by analyzing the predictive performance of combined CT-based radiomics and clinical risk factors with three models, in both enhanced and unenhanced chest CT. Their study showed that the discrimination’s predictive performance of combined, unenhanced CT-based radiomics and clinical risk factors performed better than simple radiomics models (AUCs: 0.935 vs. 0.843). Moreover, the authors found that LC patients had larger CT-size and were more likely to be older (>50 years old) than patients with granulomas, similarly to our results.

On the same line, Dennie C. and colleagues [28] reported that CT texture analysis achieved good accuracy in differentiating LC and granulomas lesions (AUC: 0.90, 88% of sensitivity, and 92% of specificity). These encouraging results may be reflecting the strong tumor heterogeneity compared to granulomas.

Future perspectives on LC and TB discrimination were provided by the promising results of Feng B. and colleagues [29], who investigated the diagnostic performance of a CT-based deep learning nomogram (DLN) in 550 patients with solitary solid pulmonary nodules. In their study, deep learning signature, gender, age, and lobulated shape, as independent predictors, were used to build the DLN; this combined model showed better diagnostic accuracy than any single model (AUCs 0.889, 0.879, and 0.809, in the training, internal validation, and external validation cohorts, respectively).

Prompt differential diagnosis between TB and LC is crucial to provide appropriate management and to avoid delayed diagnosis and treatment of patients affected by LC, which frequently leads to poor outcomes and survival [11]. In developing countries, there is limited access to invasive lung biopsy; thus, a non-invasive diagnostic method is often required. In this contest, Radiomics has emerged in the last decades, in the imaging field, as a supporting tool for clinicians in the proper management and workup of oncologic patients [20].

This study has several limitations: first, the small sample size of patients enrolled, also caused by motion artefacts on CT, could be avoided in a prospective study; secondly, the retrospective nature of the study; thirdly, intrinsic Radiomics limits, as the lack of standardization of image acquisition, lack of uniformization of image processing, operators’ subjectivity, and the lack of validation cohort.

In the future, these initial findings need to be validated, by further prospective studies, in order to overcome these drawbacks and validate Radiomics as an imaging biomarker with good reproducibility.

In conclusion, Radiomics may be a non-invasive imaging tool in many developing countries for differentiating LC from TB, and it may have a pivotal role in avoiding delayed diagnosis of LC and improving the management of oncological patients.

## Figures and Tables

**Figure 1 diagnostics-12-00739-f001:**
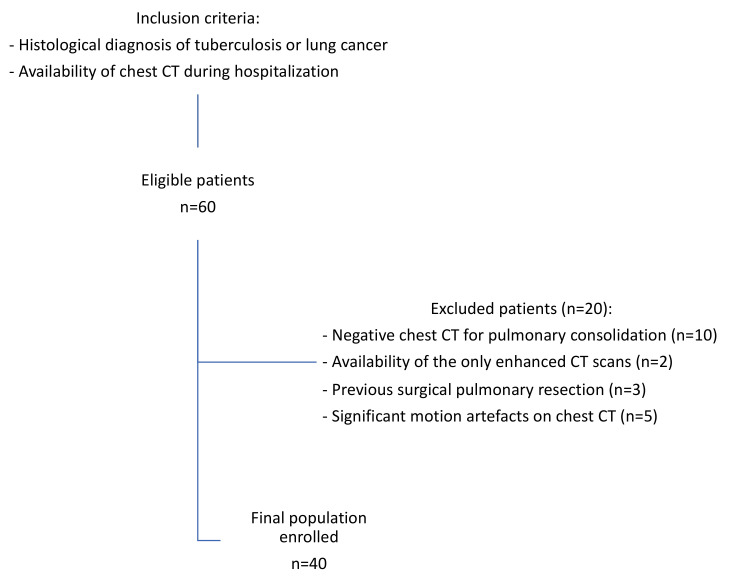
Patients’ enrollment flowchart.

**Figure 2 diagnostics-12-00739-f002:**
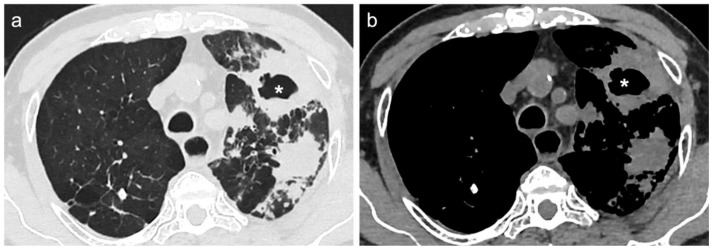
(**a**) Lung window and (**b**) mediastinal window of axial thin-section unenhanced chest CT images of a 76-year-old man, who is a former smoker. Images show multiple gross consolidation opacities without air-bronchogram, with irregular margins in the left upper lobe, as well as a cavitation (asterisks) within the contest and interstitial septal thickening. Diagnosis of lung cancer was confirmed by pulmonary biopsy.

**Figure 3 diagnostics-12-00739-f003:**
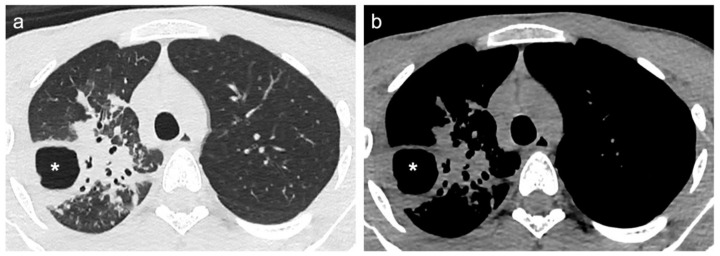
(**a**) Lung window and (**b**) mediastinal window of axial thin-section unenhanced chest CT images of a 21-year-old man. Images show a single consolidation opacity with air bronchogram and irregular margins in the right upper lobe, associated with one gross cavitation (asterisks). Diagnosis of pulmonary tuberculosis was confirmed by pulmonary biopsy.

**Figure 4 diagnostics-12-00739-f004:**
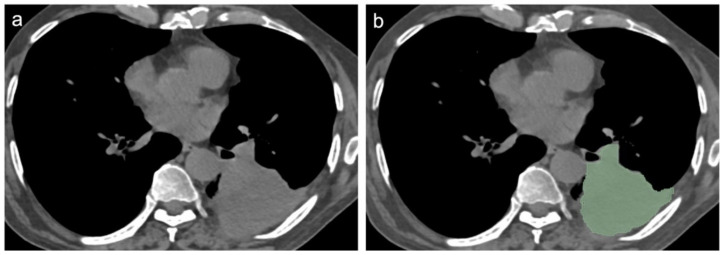
Mediastinal window of axial thin-section unenhanced chest CT images in a 67-year-old man with proven diagnosis of lung cancer. (**a**) Chest CT image shows, in the left inferior lobe, a consolidative opacity without air bronchogram. (**b**) The same image showing the consolidative opacity after radiomic volumetric segmentation (in green).

**Figure 5 diagnostics-12-00739-f005:**
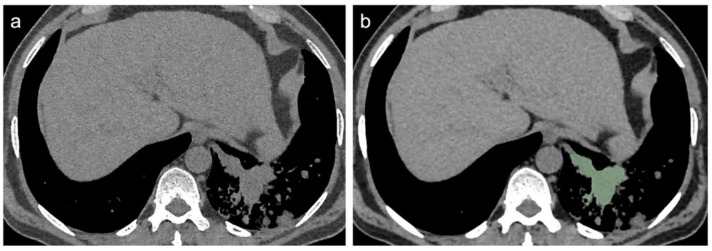
Mediastinal window of axial thin-section unenhanced chest CT images in a 84-year-old man with proven diagnosis of pulmonary tuberculosis. (**a**) Chest CT image shows, in the left inferior lobe, a consolidative opacity without air bronchogram. (**b**) The same image showing the consolidative opacity after radiomic volumetric segmentation (in green).

**Figure 6 diagnostics-12-00739-f006:**
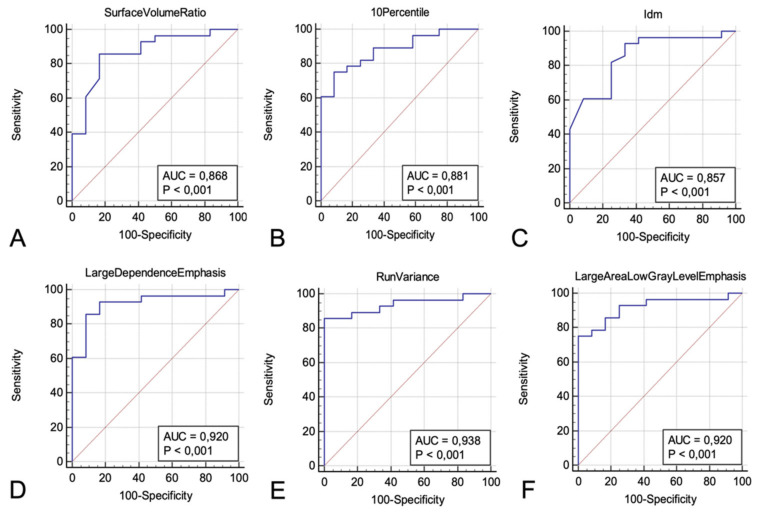
Receiver operating characteristic (ROC) curves used to test the performance of Radiomic features to discriminate lung cancer from tuberculosis, showing the best area under the curves (AUCs) for each category: (**A**) Surface Volume Ratio (AUC: 0.868, sensitivity: 85.7%, specificity: 83.3%, *p* < 0.0001) for Shape features; (**B**) 10Percentile (AUC: 0.881, sensitivity: 75%, specificity: 91.7%, *p* < 0.0001) for First Order features; (**C**) Idm (AUC: 0.857, sensitivity: 92.9%, specificity: 66.7%, *p* < 0.0001) in Grey Level Co-occurrence Matrix (GLCM) features; (**D**) Large Dependence Emphasis (AUC: 0.92, sensitivity: 85.7%, specificity: 91.7%, *p* < 0.0001) in Gray Level Dependence Matrix (GLDM) features; (**E**) Run Variance (AUC: 0.938, sensitivity: 85.7%, specificity: 100%, *p* < 0.0001) for Grey-Level Run Length Matrix (GLRLM) features; (**F**) Large Area Low Gray Level Emphasis (AUC: 0.92, sensitivity: 75%, specificity: 100%, *p* < 0.0001) for Gray Level Size Zone Matrix (GLSZM) features.

**Table 1 diagnostics-12-00739-t001:** Patients’ demographics, clinical records, and laboratory findings.

Patients Data	Total Patients (*n* = 40)	LC Patients (*n* = 28)	TB Patients (*n* = 12)	*p* Values
Mean age	59 ± 13	61 ± 8	55 ± 20	**<0.001**
Years (range)	21–82	52–77	21–82	
Male	31 (77%)	23 (82%)	8 (67%)	0.289
Female	9 (23%)	5 (21%)	4 (33%)	
Smokers	19 (48%)	16 (57%)	3 (25%)	
Non-smokers	30 (52%)	12 (43%)	9 (75%)	0.065
Comorbidities	29 (73%)	21 (75%)	8 (67%)	0.593
Blood test data				
Normal WBC count	32 (80%)	25 (89%)	7 (58%)	**0.019**
Reduced WBC count	3 (7%)	2 (7%)	1 (8%)	
Increased WBC count	5 (13%)	1 (3%)	4 (33%)	
Normal ESR	16 (40%)	14 (50%)	2 (17%)	0.052
Increased ESR	24 (60%)	14 (50%)	10 (83%)	
Normal HB	11 (28%)	21 (75%)	8 (67%)	
Reduces HB	29 (72%)	7 (25%)	4 (33%)	
Signs and symptoms				
Dyspnea	7 (18%)	6 (21%)	1 (8%)	0.445
Cough	16 (40%)	13 (46%)	3 (25%)	0.291
Fever	11 (28%)	7 (25%)	4(33%)	0.298
Weight-loss	9 (23%)	4 (14%)	5 (42%)	0.431
Asymptomatic	7 (18%)	4 (14%)	3 (25%)	0.609

Note: All significant *p* values are highlighted in bold. Normal values: WBC count 4.00–11.0 × 109/L; ESR 0–13 mm/h; HB 12–18 g/dL. Abbreviations: WBC (White Blood Cell), ESR (Erythrocyte Sedimentation Rate), HB (Hemoglobin).

**Table 2 diagnostics-12-00739-t002:** Radiomic features. * Holm–Bonferroni correction.

RADIOMIC FEATURES	LC Patients	TB Patients	
SHAPE	Mean ± SD	Mean ± SD	*p* Value *
Surface Volume Ratio	0.19 ± 0.07	0.40 ± 0.29	0.0003
FIRST ORDER			
10Percentile	−17.00 ± 98.34	−272.02 ± 365.81	0.0002
Mean	21.39 ± 28.18	−41.16 ± 77.67	0.0003
GLCM			
Difference Average	0.35 ± 0.41	1.31± 0.97	0.0004
Idm	0.90 ± 0.08	0.75 ± 0.11	0.0004
GLDM			
Large Dependence Emphasis	412.52 ± 99.93	217.46 ± 101.43	<0.0001
Small Dependence Emphasis	0.05 ± 0.03	0.13 ± 0.05	<0.0001
GLRLM			
Long Run Emphasis	26.40 ± 17.96	6.57 ± 3.61	<0.0001
Run Length Non Uniformity Normalized	0.26 ± 0.11	0.47 ± 0.15	<0.0001
Run Percentage	0.32 ± 0.12	0.57 ± 0.15	<0.0001
Run Variance	11.48 ± 7.45	2.71 ± 1.77	<0.0001
Short Run Emphasis	0.50 ± 0.12	0.70 ± 0.11	<0.0001
GLSZM			
Large Area Emphasis	101,975.94 ± 128,709.67	12,348.75 ± 14,796.78	0.0002
Large Area Low Gray Level Emphasis	5224.23 ± 21,362.92	58.69 ± 53.66	<0.0001
Zone Percentage	0.05 ± 0.04	0.15 ± 0.08	<0.0001
Zone Variance	100,830.32 ± 127,515.59	12,256.00 ± 14,766.60	0.0002

## Data Availability

The data presented in this study are available on request from the corresponding author.

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
