# Peer review of "The Role of Chest CT Radiomics in Diagnosis of Lung Cancer or Tuberculosis: A Pilot Study"

_diagnostics, 2022, doi:10.3390/diagnostics12030739_

Round 1
Reviewer 1 Report
The paper ”Lung Cancer or Tuberculosis: The Role of Chest CT Radiomics in Diagnosis of Lung Cancer” by Padmakumari and colleagues provides initial signs of potential for chest CT radiomics in differentiating lung cancer from tuberculosis. The size of this retrospective study is small (screened: 60, included: 40; 28 lung cancer, 12 tuberculosis patients), but the findings are interesting. The paper is, apart from a few minor typos, well written and easy to follow. The statistical analysis does not take multiple testing into account by which the results will be strengthened substantially; luckily, this can be easily achieved by means of, say, Holm-Bonferroni correction (see details below). The discussion should be structured into subsection, see an example below.
Specific comments
Lines 2-3: add “, a pilot study” to the title
L.27: round SD of age to 13 in order to harmonize with respective numbers in the main text and Table 1
L.28 and 167: the direction of the difference is unclear; I propose to rephrase as follows: Significant differences were found in normal WBC count (p<0.019) and age(p<0.001), in favor of the LC group (89% vs. 58%) and with an older population in LC group, respectively.
L.28-29: significant difference were NOT found in 61, but probably in 16 comparisons after adjustment for multiple testing. Please correct.
L.34: please add the following sentence to realistically relate your findings to what has been shown and what is still missing (see also comment further below, l.148-149): Future, prospective studies validating these initial findings are warranted and will have to address discrimination of TB and LC from other findings as well.
L.44: have shown (not showed)
L.59-60, 64-65, 102-103, 136-137: delete line break
L.74-75: In the recent literature, only few studies…TB from LC (e.g. [24]). (Alternatively, name several other studies.)
L.78: radiomics, not Radiomic
L.79: delete , in (comma superfluous, 2x in)
L.116: evaluated
L.134-136: Significance level and software package should be placed last in the statistics section. Please move to l.144 and rephrase like: Statistical significance was assessed with P<0.05. For inferential comparisons (Table 2), correction for multiple testing was done with Holm-Bonferroni method that is the smallest P value in Table 2 was compared to 0.05/107=0.00047 (NEW REF: Holm, S. A simple sequentially rejective multiple test procedure. Scandinavian Journal of Statistics. 1979, 6(2), 65–70). Statistical analysis was performed using MedCalc Statistical Software version 20.013 134 (MedCalc Software bvba, Ostend, Belgium).
L.143: were evaluated, too (instead of were tested too)
L.148-149: 10 out of 60 patients were excluded due to absence of TB consolidation (and probably LC as well). These are the patients in whom an algorithm in a prospective series must also perform. It is therefore essential that above limitation / perspective is added to the abstract’s conclusion, thanks.
L.150-151: motion artefacts led to the exclusion of 8% of the screened patients; can motion artefacts be avoided in a prospective study by, say, repetition of the scanning procedure. Please add to limitations of this study and as perspective to future studies in the discussion.
Table 1: header should read LC patients, not CA patients
Figs. 4 and 5: reverse order – first LC, then TB (in accordance with tables and Figs. 2 and 3)
L.214-254: correct for multiplicity and number of significant findings (probably 16, not 61). E.g. L.218-219: only Surface Volume Ratio is statistically significant different between LC and TB groups after adjustment for multiple testing.
Table 2, header: please rename Tuberculosis and Lung cancer columns to TB patients and LC patients in accordance with Table 1. Move column of LC patients to the left of TB patients to use the same structure as in Table 1.
Table 2: Root Mean Squared should probably read Root Mean Squared Error; please check and correct
Table 2: statistical significant differences between LC and TB patients and after adjustment for multiple testing were: Surface Volume Ratio; 10th Percentile; Mean; Difference Average; Idm; Large Area Emphasis; Zone Variance and all variables with P < 0.0001. Moreover, there may be even a few more as the level of significance is continuously increased in the Holm-Bonferroni method (in opposition to the classical Bonferroni method that would apply 0.05/107=0.00047 to all comparisons / p-values here). For the first comparison that is the one with the smallest p-value, the adjusted level of significance is 0.05/107=0.00047. For the second smallest p-value, it is 0.05/106=0.00047; for the third smallest, 0.05/105=0.00048 and so on until a statistically insignificant comparison is identified.
L.268-333: The discussion should be structured to ease readability and increase focus. An earlier BMJ editorial (Docherty M, Smith R. The case for structuring the discussion of scientific papers. BMJ. 1999 May 8;318(7193):1224-5. https://doi.org/10.1136/bmj.318.7193.1224) suggested the following structure for discussion of scientific papers:
- Statement of principal findings
- Strengths and weaknesses of the study
- Strengths and weaknesses in relation to other studies, discussing particularly any differences in results
- Meaning of the study: possible mechanisms and implications for clinicians or policymakers
- Unanswered questions and future research
Such a structure immediately tackles the three main questions to be answered: what were the main findings? What is new to what is known? What are the implications, what should change now?
Author Response
1) Lines 2-3: add “, a pilot study” to the title
Reply: Thanks for your valuable comment. We have modified the title adding “a pilot study” as you suggested.
2) L.27: round SD of age to 13 in order to harmonize with respective numbers in the main text and Table 1
Reply: Thank you for the punctual comment. We modify text and table as requested.
3) L.28 and 167: the direction of the difference is unclear; I propose to rephrase as follows: Significant differences were found in normal WBC count (p<0.019) and age(p<0.001), in favor of the LC group (89% vs. 58%) and with an older population in LC group, respectively.
Reply: We want to sincerely thank you for your precise and detailed comments. We have rephrased as you suggested.
4) L.28-29: significant difference were NOT found in 61, but probably in 16 comparisons after adjustment for multiple testing. Please correct.
Reply: Thank you for the comment, we have corrected the error accordingly.
5) L.34: please add the following sentence to realistically relate your findings to what has been shown and what is still missing (see also comment further below, l.148-149): Future, prospective studies validating these initial findings are warranted and will have to address discrimination of TB and LC from other findings as well.
Reply: We have added the suggested sentence, sincerely thanks for the valuable hint.
L.44: have shown (not showed)
Reply: Thanks for your correction.
L.59-60, 64-65, 102-103, 136-137: delete line break
Reply: We apologize for the errors; we have proceeded to delete the line break.
L.74-75: In the recent literature, only few studies…TB from LC (e.g. [24]). (Alternatively, name several other studies.)
Reply: Sincerely thanks for your accurate comment. We have added other references as requested.
L.78: radiomics, not Radiomic
Reply: Thanks for your correction.
L.79: delete , in (comma superfluous, 2x in)
Reply: We apologize for the typing errors that are now corrected on the Manuscript.
L.116: evaluated
Reply: We apologize for the grammatical error that we have corrected.
L.134-136: Significance level and software package should be placed last in the statistics section. Please move to l.144 and rephrase like: Statistical significance was assessed with P<0.05. For inferential comparisons (Table 2), correction for multiple testing was done with Holm-Bonferroni method that is the smallest P value in Table 2 was compared to 0.05/107=0.00047 (NEW REF: Holm, S. A simple sequentially rejective multiple test procedure. Scandinavian Journal of Statistics. 1979, 6(2), 65–70). Statistical analysis was performed using MedCalc Statistical Software version 20.013 134 (MedCalc Software bvba, Ostend, Belgium).
Reply: We want to sincerely thank you for the pivotal comments and suggestions. We have clarified that the correction for multiple testing was performed with Holm-Bonferroni method, adding the suggested reference. L.134-136 were moved and rephrased accordingly. We apologize again for the inaccuracy.
L.143: were evaluated, too (instead of were tested too)
Reply: Thank you for this correction.
L.148-149: 10 out of 60 patients were excluded due to absence of TB consolidation (and probably LC as well). These are the patients in whom an algorithm in a prospective series must also perform. It is therefore essential that above limitation / perspective is added to the abstract’s conclusion, thanks.
Reply: We extend to you our most sincere thanks for the comments. We agree with you to that is essential adding these important considerations in the abstract’s conclusion, as we have done.
L.150-151: motion artefacts led to the exclusion of 8% of the screened patients; can motion artefacts be avoided in a prospective study by, say, repetition of the scanning procedure. Please add to limitations of this study and as perspective to future studies in the discussion.
Reply: Sincerely thanks for your accurate comment. We have added this aspect in limitations paragraph. We are sorry to have not specified earlier.
Table 1: header should read LC patients, not CA patients
Reply: We are sorry for this mistake, we have corrected header in Table 1.
Figs. 4 and 5: reverse order – first LC, then TB (in accordance with tables and Figs. 2 and 3)
Reply: Thank you for the punctual comment; we have modified the figure accordingly.
L.214-254: correct for multiplicity and number of significant findings (probably 16, not 61). E.g.
Reply: Sincerely thank you for this comment. In line with the comments above, we have proceeded to clarify and correct the results according to correction for multiple testing.
L.218-219: only Surface Volume Ratio is statistically significant different between LC and TB groups after adjustment for multiple testing.
Reply: Sincerely thanks for your comment. After adjustment for multiple testing, we have corrected this sentence and the following too.
Table 2, header: please rename Tuberculosis and Lung cancer columns to TB patients and LC patients in accordance with Table 1. Move column of LC patients to the left of TB patients to use the same structure as in Table 1.
Reply: Thanks for your punctual comment. We have modified Table 1 in accordance with your comment.
Table 2: Root Mean Squared should probably read Root Mean Squared Error; please check and correct
Reply: We apologize for the inaccuracy; we have corrected it.
Table 2: statistical significant differences between LC and TB patients and after adjustment for multiple testing were: Surface Volume Ratio; 10th Percentile; Mean; Difference Average; Idm; Large Area Emphasis; Zone Variance and all variables with P < 0.0001. Moreover, there may be even a few more as the level of significance is continuously increased in the Holm-Bonferroni method (in opposition to the classical Bonferroni method that would apply 0.05/107=0.00047 to all comparisons / p-values here). For the first comparison that is the one with the smallest p-value, the adjusted level of significance is 0.05/107=0.00047. For the second smallest p-value, it is 0.05/106=0.00047; for the third smallest, 0.05/105=0.00048 and so on until a statistically insignificant comparison is identified.
Reply: We want extend to you our most sincere thanks for the comment and the punctual explanation. According to your suggestion, we have modified Table 2 leaving all the 16 significative features after Holm-Bonferroni correction. For the same reason, we have to modified Figure 6 too, cutting off “contrast” feature’s image.
L.268-333: The discussion should be structured to ease readability and increase focus. An earlier BMJ editorial (Docherty M, Smith R. The case for structuring the discussion of scientific papers. BMJ. 1999 May 8;318(7193):1224-5. https://doi.org/10.1136/bmj.318.7193.1224) suggested the following structure for discussion of scientific papers:
- Statement of principal findings
- Strengths and weaknesses of the study
- Strengths and weaknesses in relation to other studies, discussing particularly any differences in results
- Meaning of the study: possible mechanisms and implications for clinicians or policymakers
- Unanswered questions and future research
Such a structure immediately tackles the three main questions to be answered: what were the main findings? What is new to what is known? What are the implications, what should change now?
Reply: We extend to you our most sincere thanks for the valuable and punctual comments. We have rewritten and structured the Discussion section according to BMJ editorial that you have suggested.
Reviewer 2 Report
While the general idea of the paper has some scientific value if correctly investigated and implemented, the paper has several shortcomings:
(1) In Methods the authors mention that 107 radiomic features were extracted however without offering any information on how the radiomic features were chosen.
(2) radiomic analysis based on 40 patients cannot offer a good statistical power. The number of patients seems inadequate for such study thus the conclusions are not representative for a large group of patients. Even the studies mentioned in Discussion include hundreds of patients for radiomic analysis.
(3) the results obtained from the retrospective cohort were not validated – this is a strong shortcoming of the study.
(4) the limitations section must include a paragraph on the boundaries of radiomics that are linked to several aspects: the lack of standardisation of image acquisition, lack of uniformization of image processing, operators’ subjectivity – just to name a few. This is particularly important if images acquired at different centres are used for radiomic analysis.
Author Response
1) In Methods the authors mention that 107 radiomic features were extracted however without offering any information on how the radiomic features were chosen.
Reply: Sincerely thanks for your valuable comment. The volumetric lung segmentation was performed by using the open-source 3D Slicer software; the software itself extracted routinely 107 radiomic features mentioned. We have specified this aspect in 3D segmentation and radiomic features subsection of Results. We hope that our explanations have better clarified your concerns.
2) radiomic analysis based on 40 patients cannot offer a good statistical power. The number of patients seems inadequate for such study thus the conclusions are not representative for a large group of patients. Even the studies mentioned in Discussion include hundreds of patients for radiomic analysis.
Reply: We want to sincerely thank you to have noted this aspect and for your punctual comment. We are aware that the current study has enrolled a small population sample. In line with the comments above, we have proceeded to perform the correction of the results for multiple testing through Holm-Bonferroni method, in order to offer better statistical power to our results. As a matter of fact, our future aim will be performing a prospective study to discriminate between TB and LC by Radiomics, enrolled only patients with similar pulmonary consolidations at chest CT, in order to obtain more homogeneous and powerful results.
3) the results obtained from the retrospective cohort were not validated – this is a strong shortcoming of the study.
Reply: Many thanks for your punctual comment. We are conscious of this important shortcoming, and we have underlined it in “limitations” paragraph of Discussion section. As affirmed above, our future aim will be performing a prospective study with an internal and external validation too. Moreover, we have changed the title in “The Role of Chest CT Radiomics in Diagnosis of Lung Cancer or Tuberculosis: a pilot study” and we have clarified that further future studies will be necessary to validate these initial findings.
4) the limitations section must include a paragraph on the boundaries of radiomics that are linked to several aspects: the lack of standardisation of image acquisition, lack of uniformization of image processing, operators’ subjectivity – just to name a few. This is particularly important if images acquired at different centres are used for radiomic analysis.
Reply: Sincerely thank you for your valuable and punctual observations. We agreed with your comment, and we have modified Discussion section adding these important limitations, as you kindly have suggested. However, we want to precise that the CT scans were acquired in only one center.
Round 2
Reviewer 1 Report
Thank you for providing a major revision that incorpororated many of my earlier suggestions. Appreciated. I think the paper now is more transparent about what has been achieved and what still needs to be pursued. Best of luck with your future, prospective endeavors. Please correct the following minor typographic misspellings during the proof-reading phase, thanks.
L.137 standard deviation, not Standard Deviation
L.360, 372 delete commas before that
L.376 need to be validated (not validating)
L.447 reference 27 lacks volume and page information and should read as follows: Holm, S. A simple sequentially rejective multiple test procedure. Scandinavian Journal of Statistics. 1979, 6(2), 65–70.
Reviewer 2 Report
The authors have addressed all comments raised by this reviewer. The change of title is more illustrative of the content of the paper. The article has overall improved.